# Prevalence of sexual violence and its associated factors among housemaids attending evening schools in urban settings of Gedeo zone, Southern Ethiopia: A school based cross sectional study

**Kalkidan Gezahegn, Selamawit Semagn, Mohammed Feyisso Shaka**[ID] *

School of Public Health, College of Health Sciences and Medicine, Dilla University, Dilla, Ethiopia

* mohammedshaka2@gmail.com

## Abstract

### Background

Housemaids are the most vulnerable group to sexual violence due to their working condition, isolation and school arrangements. Despite the ubiquity of sexual violence among the domestic work sector, particularly among housemaids, this area of research continues to be neglected. This study was aimed at examining the prevalence and factors pertaining to sexual violence among housemaids attending night school program in urban setups of Gedeo zone, Southern Ethiopia.

### Method

A school based quantitative cross sectional study was conducted among 394 housemaids attending night schools in the urban setups of Gedeo Zone from April to May 2019. After stratifying of students using class grade, SRS technique was used to choose study subjects from each stratum. Quantitative data was collected using face to face interview and qualitative was collected using focus group discussion, in-depth interview and key informant interview. The data was entered and analyzed by SPSS version 20. Binary logistic regression was fitted to determine the association of each independent variable with the dependent variable.

### Result

Based on the finding, the prevalence of sexual violence was 60.2%. The odds of experiencing sexual violence through working life-time as housemaid was higher for those who had migrated from rural to urban for work [AOR = 1.97: 95% CI, (1.07,3.63)], had less than 5 years of experience as housemaid [AOR = 3.10: 95% CI, (1.60, 6.00)], were in the age group of 15–19 [AOR = 3.75:95% CI (1.88, 7.46), ever used alcohol [AOR = 6.77: 95% CI, (2.65,17.33)] and whose fathers lacked formal education [AOR = 2.75:95%CI (1.24,6.08)]. On the other hand, unmarried /housemaids having no regular sexual partner were less likely to face sexual violence [AOR. = 0.28: 95% CI, (0.13, 0.57)].

**Data Availability Statement:** Data underlying this study is readily available in Dryad public data repository: https://datadryad.org/stash/share/

XNXalbtgvA7-
j18FOd5lzZ57w3Tap2PurjhQIAMxwyA.

**Funding:** Funding for this study was covered by Dilla University NORHED/SENUPH project and the funding organization have no any role in methodology, analysis, interpretation, and write up of the manuscript.

**Competing interests:** The authors have declared that no competing interests exist.

## Conclusion

The level of sexual violence was found to be high among housemaids attending night schools in in this study. Housemaids from rural area, those newly starting the work, younger housemaids, and those who were married were more likely to be victims of sexual violence.

## Background

Sexual violence is defined as: 'Any sexual act, attempt to obtain a sexual act, unwanted sexual comments or advances, or acts to traffic or otherwise directed against a person's sexuality using coercion, by any person regardless of their relationship to the victim, in any setting, including but not limited to home and work. It can take many forms including a threat of rape, attempted rape, complete rape, sexual harassment and sexual contact with force [1]. In any of the forms of sexual violence, minority and marginalized women are the most vulnerable, and generally those who face the greatest obstacles to gain protection and necessary services [2].

Domestic work is an occupation for millions of women worldwide. According to international labor organization (IOL) 2010 estimates, domestic workers represent 4–10% and 1–1.5% of the total workforce in developing and developed countries respectively [3]. In Ethiopia, there is a very high prevalence of seasonal and long-term rural-urban migration for domestic work. The main destinations of these migrants are majorly regional and zonal towns, which provide these migrants with domestic services as a housemaid [4]. Domestic work is one of the least protected sector under labor law and poor monitoring and implementation of the existing laws puts housemaids in a highly disadvantaged position that in turn exacerbates their vulnerability to sexual abuse and violence [5, 6].

According to evidences from different literatures, the sexual activity of housemaid women also differs from that of the general population. A population based study conducted in urban areas of Ethiopia revealed that, domestic workers were more likely to have had sex before age 15 and to have been coerced into having sex when compared to other young women [7].

Housemaids often comes across different types of sexual abuse ranging from verbal harassment to forced sex by male members of the household, brokers and friends [5, 7–9]. Survival becomes primary consideration for the victims because they often have few or no options for other work, and they also lack awareness and knowledge of their options since most come from poor families, are migrants from the rural areas, and have low level of education. As a result, they usually cannot report such offences so that these kinds of violence against house maids remain usually concealed [9–11].

Most housemaids do not get the opportunity to continue their education on a regular program. Consequently, they are obliged to attend the night shift, which increases their vulnerability to sexual abuse. Many of the studies related to youths conducted in the context of regular secondary schools and higher educational institutions with an emphasis on sexual violence, with very few studies carried out on the most vulnerable group in the evening program [12–14].

Despite the number of studies revealing the vulnerability of housemaids to sexual abuse and exploitation, there is a substantial lack of data on the magnitude of sexual violence among housemaids in Ethiopia.

## Methods and materials

### Study design and setting

A school based quantitative cross sectional study was conducted in urban settings of Gedeo zone, Southern Ethiopia, from April 1 to May 1, 2019. Gedeo Zone is located at 369 km to the

South of Addis Ababa. This study was conducted in the two urban administrative (Dilla and Yirgachefie towns) in the district. The study was conducted in five schools (1 high school & four primary schools) in the two town administrative.

## Population and sample size

Female housemaids aged 15 years and above and attending evening schools during the study period in the districts were included. Sample size was determined using single population proportion formula considering the following assumptions: prevalence of sexual violence among housemaids assumed to be 50% due to lack of comparable previous study, 95% certainty and maximum discrepancy of 5% between the sample size and the underlining population. The adopted sample size formula is:

$$n = (Z\alpha/2)^2 p(1-p)/d^2 \tag{1}$$

Where:
n = sample size
P = prevalence of sexual violence among house maids = 50%
d = margin of error 5%
$Z\alpha/2$ = critical value at 95% confidence level of certainty (1.96)
Accordingly, the calculated sample size, n = $(1.96)^2$ x 0.5(1–0.5) / $(0.05)^2$ = 384
Finally 10% of non-response rate was added making the total sample size of this study 422.

## Sampling technique

Among the schools having evening program, five schools were randomly selected and census was conducted to identify students who were housemaids. To identify the housemaids, all students attending selected evening schools were asked to fill a form on the type of job they do. Different codes were assigned to different possible types of occupations along with their ages and class role number to identify eligible respondents. Then the total number of housemaid students in each of evening schools were identified with their respective class grades and the sample size was proportionally allocated to each class. In this regard, the distribution of total number of housemaid students in each class were:

- 45 students in grade 9 (no housemaid student was obtained with educational level above grade nine)

- 166 students in grade 7 and 8

- 122 students in grade 5 and 6

- 124 students in grade 3 and 4

- 150 students in grade 1 and 2

With proportional allocation to each size of the total students in each classes, the allotted number of students for each category was:

- 31 students in grade 9

- 116 students in grade 7 and 8

- 85 students in grade 5 and 6

- 86 students in grade 3 and 4

- 104 students in grade 1 and 2

Then simple random sampling technique was used to choose the proportionally allocated study participants from each class. For eligible participants who were absent at the day of data collection, data collectors were revisited the class room two times at different time intervals and for those who were absent after two revisits, the participant was replaced by another in the sampling frame.

## Data collection procedure and data quality control

Data was collected with face to face interview by six data collectors using structured questionnaire. To reduce the possibility of bias and to make participants respond genuinely, specialized training was adequately provided for data collectors and interview was conducted at isolated place with adequate privacy. Confidentiality of the participants was highly protected. The instrument was adapted from previous published literatures and WHO multi-country study on women's health [15–17] and it was modified depending on the local situation and the research objective. The tool contains socio demographic characteristics of respondents, family and employers characteristics, behavioral information and sexual history. The questionnaire was prepared in English, and translated to Amharic to use in the field. The tool was also pretested before the actual use in the field. Pre-test was done using 10 house maids attending night schools in Wonago town which is the nearby district to the actual data collection site, with comparable socio-demographic and socio-economic characteristics.

Data collection process was supervised by the investigators and employed supervisors for this purpose. Training was given for data collectors and supervisors to orient them on the objectives and ethical conduct of the study, the nature of the study, the sensitiveness of the issue and how to minimize any bias during interview.

## Measurement

Sexual violence experience for each housemaid was defined if a housemaid faces one or more of the following: rape, attempted rape, coerced sex, any sexual harassment at list once in her life-time course as housemaid.

**Sexual harassment.**   When the participant experienced one of the following: unwanted and repeated sexual advances that range from unwelcome comments and improper or offensive verbal jokes, kissing and touching her sensitive body parts.

Problematic substance use or involvement was measured using ASSIST tool. From the measurement, a score of greater than 5 was used for alcoholic beverages (like areke, tela, beer, etc.) and for any other type of substance (shisha, chat, etc.) a score of greater than 2 was considered as problematic substance use or involvement [18].

## Data processing and analysis

The quantitative data was checked, coded and entered to Epi-Data version 3.1 and was exported to SPSS (Statistical Package for Social science) version 20 for analysis. In descriptive statistics tables, graphs, mean and frequency was used to present the finding. Binary logistic regression was fitted to identify factors associated with sexual violence. The strength of statistical association between dependent and independent variables was measured at p-value<0.05 by adjusted odds ratio with their corresponding 95% confidence interval.

## Ethical consideration

Ethical clearance was obtained from Institutional Review Board of Dilla University, College of Health Science and Medicine. Written informed consent was obtained from each respondent

after detailed explanation of the nature and procedure of the study. Written consent was obtained from the non-minor respondents before conduction of the study. For minors, the assent was obtained from their parents/guardian for those whom their parents/guardian were accessible. But for some minors whom their parents/guardians were not accessible due to different reasons, the IRB was clearly informed about the problem and their assent was waived. The information given by each respondent was kept with strict confidentiality and name was not recorded.

## Result

### Socio-demographic characteristics of the housemaids

Out of the 422 housemaids that were expected to participate in this study, a total of 394 were interviewed allowing a response rate of 93.4%. It was noted that, the mean age of the study participants was 18 years (SD ±2.5) with the minimum age being 15 years and maximum age being 24 years. Among these, 279 participants (70.8%) were adolescents (aged 15–19 years), whereas 115 (29.2%) were young adults (aged 20–24 years). A majority of the study's participants 316 (80.2%) were neither married nor had a boyfriend. Regarding their educational status, it was found that 264 (67%) participants, i.e. large number of them, were educated only up to the 6th grade or below. In all 277 (70.3%), of the respondents earned between 100–500 birr as monthly salary and an almost equal number of them, i.e., 294 (74.6%) had no family support. As for the family characteristics of the participants, the parents of 135 (34.3%) of them were either divorced or one of the parent had passed away. Educationally, 159 (53.4%) of the respondents' fathers and 251 (63.7%) of the respondents' mothers had no formal education. Out of the total participants, 346 (87.8%) were from families with low income (Table 1).

### Employment and employer characteristics of the housemaid

A majority of the housemaids, 332 (84.3%), were live-in domestic workers and the remaining 62 (15.7%) of them were live-out workers. The respondents were asked about their residence before coming to the present place of residence. About 288 (73.1%) of them were from rural areas. Regarding their work experience as housemaids, 286 (72.6%) had worked for more than five years and the mean duration of their work experience was four years with a minimum of one year and a maximum of ten years.

Some information about their employers was also asked of the housemaids who participated in this study. Among the employers, almost all of them were married and 232 (58.9%) had attended higher education (Table 2).

**Sexual history and behavioral characteristics.** Among the total study participants, 103 individuals (26.1%) had a life time history of sexual intercourse, and of those individuals, 55 (53.4%) were sexually active during the past 12 months. The mean age at first sexual intercourse was 17, and for the largest number of the sexually actives, the reason for initiation of sexual activity was forced sex (39 individuals, 37.9%), followed by personal desire (24 individuals, 23.3%). On the other hand, 53 (13.6%) had ever used alcohol (Table 3).

Life-time substance use and current (within the past 3 months) substance use were assessed using the ASSIST tool. Based on the scoring of the tool, none of the participants fulfilled the criteria to be classified as substance users. The study showed that, 53 (13.3%) of the study participants had ever used alcohol in their life-time, and of that group 45 consumed alcohol once or twice in the past 3 months. Only 8 students (2.0%) had ever chewed chat, and only 2 students (0.5%) had ever smoked shisha. None of the respondents had ever smoked a cigarette.

**Table 1. Socio demographic and family characteristic of housemaids attending night schools in urban settings of Gedeo zone southern Ethiopia, May 2019.**

| Variable | Frequency | Percent |
|---|---|---|
| Age category | 394 | 100 |
| 15–19 | 279 | 70.8 |
| 20–24 | 115 | 29.2 |
| Mean | 18± 2.5 | |
| Religion of the respondent | 394 | 100 |
| Orthodox | 118 | 29.9 |
| Protestant | 257 | 65.2 |
| Muslim | 12 | 3.2 |
| Catholic | 7 | 1.8 |
| Marital status of the respondents | 394 | 100 |
| Never married | 308 | 78.1 |
| Regular boyfriend | 60 | 15.22 |
| Married | 18 | 4.5 |
| Divorced/ Widowed | 8 | 2.02 |
| Monthly salary | 394 | 100 |
| ≤100 | 57 | 14.5 |
| >100–500 | 277 | 70.3 |
| >500 | 60 | 15.2 |
| Educational level | 394 | 100 |
| First cycle (grade 1–4) | 182 | 46.2 |
| Second cycle (grade 5–8) | 188 | 47.7 |
| High school (grade 9) | 24 | 6.1 |
| Child hood residence | 394 | 100 |
| Rural | 288 | 73.1 |
| Urban | 106 | 26.9 |
| Father educational status | 298 | 100 |
| No formal education | 159 | 53.4 |
| Primary education | 98 | 32.9 |
| Secondary education and above | 41 | 13.8 |
| Mother educational status | 332 | 100 |
| No formal education | 251 | 75.6 |
| Primary education | 61 | 18.4 |
| Secondary education and above | 20 | 6.0 |
| Family income | 394 | 100 |
| Medium | 48 | 12.2 |
| Poor | 346 | 87.8 |
| Family support | 394 | 100 |
| Yes | 100 | 25.4 |
| No | 294 | 74.6 |

## Prevalence of sexual violence among housemaids

The prevalence of life time sexual violence among the housemaids in this study was 237 (60.2%). The most prevalent type of sexual violence was sexual harassment, reported by 230 (58.4%) of the respondents. The prevalence of completed rape and attempted rape in life-time were 44 (11.2%) and 75 (19%) respectively (Fig 1). Regarding the perpetuators of the violence,

**Table 2. Employment and employer characteristics of housemaids attending evening schools in urban settings of Gedeo zone Southern Ethiopia, May 2019.**

| Variable | Frequency | Percent |
|---|---|---|
| Agreement type | **394** | **100** |
| Live-in | 332 | 84.3 |
| Live-out | 62 | 15.7 |
| Duration of residence in the town | 394 | 100 |
| Five years and below | 217 | 55.1 |
| Above five years | 177 | 44.9 |
| Experience as a housemaid | **394** | **100** |
| Five years and below | 286 | 72.6 |
| Above five years | 108 | 27.4 |
| Employer marital status | **394** | **100** |
| Married | 372 | 94.4 |
| Single | 22 | 5.6 |
| Occupation of employer | **394** | **100** |
| Merchant | 100 | 25.4 |
| Government employee | 245 | 62.2 |
| Private | 47 | 11.9 |
| Other | 2 | 0.5 |
| Family size of the employer | **394** | **100** |
| Less than three | 30 | 7.6 |
| Three to five | 166 | 42.1 |
| Greater than five | 198 | 50.3 |

**Table 3. Sexual history of housemaids attending night schools in urban settings of Gedeo zone southern Ethiopia, May 2019.**

| Variable | Frequency | Percent |
|---|---|---|
| Sexual debut | 394 | 100 |
| Yes | 103 | 26.1 |
| No | 291 | 73.9 |
| Age at first sex in years | 103 | 100 |
| 10–14 | 12 | 11.7 |
| 15–19 | 79 | 76.7 |
| $\geq$20 | 12 | 11.7 |
| Reason for first sex | 103 | 100 |
| Forced | 39 | 37.9 |
| Desire | 24 | 23.3 |
| Marriage | 18 | 17.5 |
| Peer pressure | 15 | 14.6 |
| For money or promising word | 7 | 6.8 |
| Life time regular sexual partner | 394 | 100 |
| None | 336 | 85.3 |
| One | 39 | 9.9 |
| Two and above | 19 | 4.8 |
| Sex in the last 12 months | 394 | 100 |
| Yes | 55 | 14 |
| No | 339 | 86 |

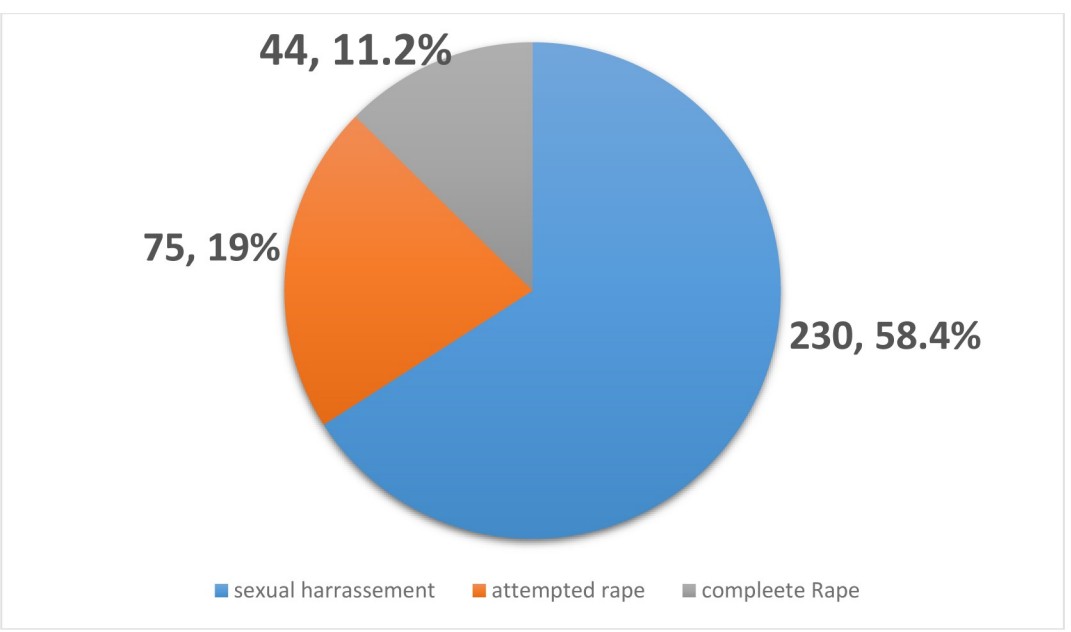

**Fig 1. Prevalence of different forms of life time sexual violence among housemaids attending night schools in urban setups of Gedeo zone, May 2019.** NB. Respondent can report more than one form of sexual violence.

75 instances (31.6%) were performed by a person unknown by the victim and 54 (22.8%) were perpetrated by the broker or other intermediary person (Table 4).

### Factors associated with sexual violence among housemaids

After adjustment for possible confounders on multivariate analysis; age of the housemaid, childhood place of residence, duration of experience as a housemaid, relationship status of the housemaids, ever alcohol use by a housemaid and father's educational level were identified to be significantly associated with life time sexual violence.

The result of multivariable logistic regression analysis revealed that, housemaids aged 15–19 years were about four times [AOR: 3.75, 95% CI (1.88, 7.46] more likely to face sexual violence compared to those greater than or equal to 20 years. The odds of sexual violence was increased by 97% [AOR: 1.97, CI: (1.07, 3.63) for housemaids coming from rural area compared to those from urban. Regarding the year of experience as housemaid, those who worked for less than or equal to five years were three times [AOR: 3.10, CI: (1.60, 6.00)] more likely to be sexually violated compared to those who worked for more than five years.

Among the family characteristics, the odds of life time sexual violence was almost three times [AOR: 2.75, CI: (1.24, 6.08)] for those whose fathers had no formal education compared to those whose fathers educational level was secondary and above. Among the behavioral characteristics, the odds of life time sexual violence was about seven times [AOR: 6.77, CI: (2.65, 17.33)] higher among housemaids who ever used alcoholic beverages compared to those who never used any kind of alcoholic beverages.

On the other hand, housemaids who were not in a union were less likely to face sexual violence. In this regard, the likelihood of experiencing sexual violence for those who were not in union was decreased by 72% [AOR: 0.28, CI (0.13, 0.57)] compared to those who were in a union (Table 5).

**Table 4. Prevalence of sexual violence among night school housemaid students in urban settings of Geode zone southern Ethiopia, May 2019.**

| Sexual violence | Frequency | Percent |
|---|---|---|
| Life-time* sexual violence | **394** | **100** |
| Yes | 237 | 60.2 |
| No | 157 | 39.8 |
| Perpetuator | 237 | 100 |
| Employer | 38 | 16.0 |
| Employer neighbor | 16 | 6.8 |
| Employer close relative | 27 | 11.4 |
| Broker or other intermediary | 54 | 22.8 |
| Friend | 16 | 6.8 |
| Group | 11 | 4.6 |
| 75 () | | |
| Unknown person | 75 | 31.6 |
| Age of the perpetrator | 237 | 100 |
| Same age | 11 | 4.6 |
| Older than me | 81 | 34.2 |
| Much older than me | 146 | 61.6 |
| Place of occurrence of the event | 237 | 100 |
| Employer home | 65 | 27.3 |
| School compound | 38 | 15.9 |
| Jungle | 38 | 15.9 |
| Hotel | 5 | 2.3 |
| Friends home | 10 | 4.5 |
| Perpetuators home | 81 | 34.1 |

*Lifetime is a period during the respondents' life as a housemaid

## Discussion

This study assessed the prevalence of sexual violence and associated phenomena among housemaids attending evening schools in urban settings in the Gedeo Zone. The results indicate that the lifetime prevalence of sexual violence among the housemaids is 60.2% (95% CI: 55.3%–64.5%). Notably, 52.8% of the incidents occurred within the past year. The level of sexual violence among the subjects of this study represents a significant public health concern, and it reflects similar findings in a study conducted in Hosaena, in which 57.75% of the housemaids had faced sexual violence [19]. An even greater incidence has been found in studies conducted in larger cities in Ethiopia, including Harar (72%) [8], Addis Ababa (76%) [20], and Mekele (84%) [21]. This discrepancy mainly reflects the different contexts of the studies, in which there were larger numbers of the target population in various Ethiopian cities with different cultural and socioeconomic characteristics.

The level of sexual violence among housemaids in Ethiopia was much higher than the findings of 29% in India [22], 26% in Brazil [23], and 14% in Portugal [24]. Research into the nature of gender-based violence faced by domestic workers in their workplaces in three South African provinces has also uncovered shocking levels of abuse and sexual harassment by employers, with few cases ever reported [25]. Furthermore, it is usually difficult to know the true dimensions and levels of sexual violence and there is a high possibility of under-reporting associated with the deep social stigma attached to sexual assault [26, 27]. This is typically

**Table 5. Bivariate and multivariable logistic regression analysis result of the factors associated with sexual violence among housemaids attending night schools in urban settings of Gedeo zone Southern Ethiopia May 2019.**

| Variable | Sexual violence | | COR(95%CI) | AOR (95%CI) | P value |
|---|---|---|---|---|---|
| | **Yes** | **No** | | | |
| | **Frequency (%)** | **Frequency (%)** | | | |
| Age category | 237 | 157 | | | |
| 15- 19years | 188(67.4) | 91(32.6) | 2.78(1.78,4.34) | 3.75(1.88,7.46) | <0.001 |
| 20–24 years | 49(42.6) | 66(57.4) | 1 | 1 | |
| Child hood residence | 237 | 157 | | | |
| Rural | 183(63.5) | 105(36.5) | 1.67(1.07, 2.63) | 1.97(1.07,3.63) | 0.029 |
| Urban | 54(50.9) | 52(49.1) | 1 | 1 | |
| Monthly salary (Ethiopian Birr) | 237 | 157 | | | |
| ≤100 | 45(78.9) | 12(21.1) | 2.86(1.26, 6.48) | 1.88(0.66, 5.39) | 0.234 |
| 100–500 | 158(57) | 119(43) | 1.01(0.57, 1.78) | 0. 72(0.32, 1.62) | 0.435 |
| >500 | 34(56.7) | 26(43.3) | 1 | 1 | |
| Marital status | 237 | 157 | | | |
| Not in relationship | 179(56.6) | 137(43.4) | 0.45(0.25, 0.78) | .28(0.13,0.57) | 0.001 |
| In a relation ship | 58(74.4) | 20(25.6) | 1 | 1 | |
| Experience as a housemaid | 237 | 157 | | | |
| Five years and below | 195(68.2) | 91(31.8) | 3.36(2.12,5.33) | 3.10 (1.60,6.00) | 0.001 |
| Above five years | 42(38.9) | 66(61.1) | 1 | 1 | |
| Duration of stay in the present town | 237 | 157 | | | |
| Five years and below | 146(67.3) | 71(32.7) | 1.94(1.29,2.92) | 0.79(0.41,1.54) | 0.499 |
| Above five years | 91(51.4) | 86(48.6) | 1 | 1 | |
| Alcohol ever use | 237 | 157 | | | |
| Yes | 46(86.8) | 7(13.2) | 5.16(2.26,11.75) | 6.77(2.65,17.33) | <0.001 |
| No | 191(56.0) | 150(44) | 1 | 1 | |
| Father education | 237 | 157 | | | |
| No formal education | 106(66.7) | 53(33.3) | 2.31(1.15, 4.64) | 2.75(1.24,6.08) | 0.012 |
| Primary education | 54(55.1) | 44(44.9) | 1.42(0.68,2.95) | 1.38(0.55,3.44) | 0.480 |
| Secondary and above | 19(46.3) | 22(53.7) | 1 | 1 | |
| Family support | 237 | 157 | | | |
| Yes | 51(51.0) | 49(49) | 1 | 1 | |
| No | 186(63.3) | 108(36.7) | 1.65(1.04,2.61) | 1.56(0.91,2.69) | 0.106 |

happen among most of the victims of sexual violence like housemaids implying the intensive need for public attention.

The most common type of sexual violence was sexual harassment, which was reported by 230 (58.4%) of the study's participants, supporting the findings of the study in Hosaena [19]. This demonstrates that harassment can occur in numerous ways in diverse social settings. The lifetime prevalence of completed and attempted rape among the study participants was 11.2% and 19.0%, respectively, and the findings of studies from Hawassa [28] and Rwanda [29] reveal comparable levels of rape and attempted rape. Smaller percentages of such incidents were reported in studies from Harar [8], Addis Ababa [17, 20] and Mekelle [21]. The extent of diverse types of sexual violence among the study participants in all these contexts and settings indicates a need for broad action to end violence against women, which remains a considerable obstacle to the achievement of women's and girls' human rights and the 2030 Agenda for Sustainable Development.

Another interesting finding regarding sexual violence among housemaids concerns the condition of their first sexual contact. In this study, 26.1% of the housemaid students had been sexually active, and 37.9% of them were coerced into their first sexual act, revealing the severity of a problem that results in a significant number of housemaids being forced into sex before they intend to do so. This finding is consistent with that of a study conducted in three low-income urban areas in Ethiopia, in which 30.3% of sexually active housemaids described their first sexual experience as occurring under coercive conditions [7].

This study reveals that younger housemaids were more likely to be sexually violated than older ones (above 20 years), a finding consistent with that of the study from Rwanda [29]. This finding is congruent to the finding from Kenya showing being aged less than 18 years as one of the factors associated with higher experience of sexual violence [30]. Younger housemaids are physically as well as psychologically immature and thus more easily manipulated; they may have insufficient experience to enforce clear boundaries for sexual activities and may be less able to object to unwanted advances. Moreover, their lack of maturity may hinder their recognizing of warning signs early, leading perpetrators to more readily target younger housemaids who offer little or no resistance to their sexual predation.

Alcohol use was another factor associated with sexual violence among housemaids. The results show that housemaids who had ever used alcohol were almost seven times more likely to be sexually violated than those who had never drunk alcohol. This finding is in line with the studies conducted among housemaids in Hawassa [28, 31] and Addis Ababa [17]. Similar finding was reported from WHO report on understanding and addressing violence against women [32]. Apparently, consuming alcohol makes it more difficult for housemaids to protect themselves by interpreting and effectively responding to warning signs.

In this study, involvement in a relationship was also significantly associated with lifetime sexual violence. Housemaids who were not in a union or married were 72% less likely to face sexual violence than those who were either married or in a union, suggesting that the main perpetrators of sexual violence were intimate partners. This is also congruent with other studies showing similar condition in sub-Saharan Africa including Ethiopia, where common perpetrators for sexual violence among the women were intimate partner [30, 33, 34]. Housemaids with a boyfriend or husband may be less likely to resist attempts at sexual violence for fear that doing so would end the relationship.

The study participants' original area of residence was also significantly associated with lifetime sexual violence; housemaids from rural areas were almost twice as likely to be sexually violated as those from urban areas. Similarly, from study conducted in Northern part of Ethiopia housemaids who previously lived in rural areas were about three times more likely to face violence [35]. This is particularly due to the fact that housemaids from rural areas are less exposed to sexual and reproductive health information because media are not easily accessible in those areas. Additionally, they experience cultural alienation when they migrate to urban areas and may not speak the local language, causing them to face cultural and communication barriers that make them more vulnerable to sexual violence than housemaids from urban areas.

Regarding years of experience as a housemaid, those who had worked for less than five years were three times more likely to be sexually violated than those who had worked for more than five years. This is explained by the fact that housemaids who face any kind of sexual violence are less likely to remain in the sector and more likely to seek different employment to escape an abusive work environment. This conclusion is supported by the finding of a study conducted among out-of-school girls in six Ethiopian regions by the population council [7]. Additionally, housemaids who face sexual violence may be fired by female or male employers to conceal the matter and protect the employer's reputation, reducing the likelihood of a

housemaid's remaining in the sector. Housemaids who do not face sexual violence are more likely to continue working as a housemaid and may have sufficient life experience to protect themselves from being sexually violated.

## Strength and limitation of the study

This study was conducted to address the marginalized, vulnerable and the most neglected group of population with respect to sexual violence. However, important limitations were admitted in this study. The main limitation of this study was confinement to school attending housemaids which might not reflect the reality among the general population of all housemaids found in the sector. There was also serious limitation of literatures in the area to make more conclusive discussion.

## Conclusion and recommendation

In the current study, sexual violence was found to be a major problem among going housemaids. Nearly two-third of the participants have faced at least one form of sexual violence. The life-time completed rape and attempted rape level were also substantially considerable. Younger ages, housemaids who are in union, less experienced housemaids, and those residing in rural area prior to the present place of work were more likely to experience sexual violence in this study. Thus, pragmatic intervention programs aiming to solve the problem of such vulnerable segment of populations needs to be in place. Government should focus this group particularly, with respect to labor laws and regulations to insure their basic human right. Further evidence generating studies focusing on effective interventions modalities to address this population should be taken in to consideration by researchers.

## Acknowledgments

We would like to acknowledge NORHED project for financial coverage Gedeo zone for their cooperation during conduction of the project. We are also very grateful to study participants and data collectors participated in the study.

## Author Contributions

**Conceptualization:** Kalkidan Gezahegn, Mohammed Feyisso Shaka.

**Data curation:** Kalkidan Gezahegn, Mohammed Feyisso Shaka.

**Formal analysis:** Kalkidan Gezahegn, Selamawit Semagn, Mohammed Feyisso Shaka.

**Funding acquisition:** Kalkidan Gezahegn.

**Investigation:** Kalkidan Gezahegn, Selamawit Semagn, Mohammed Feyisso Shaka.

**Methodology:** Kalkidan Gezahegn, Selamawit Semagn, Mohammed Feyisso Shaka.

**Project administration:** Kalkidan Gezahegn, Selamawit Semagn, Mohammed Feyisso Shaka.

**Resources:** Kalkidan Gezahegn, Mohammed Feyisso Shaka.

**Software:** Kalkidan Gezahegn, Selamawit Semagn, Mohammed Feyisso Shaka.

**Supervision:** Kalkidan Gezahegn, Selamawit Semagn, Mohammed Feyisso Shaka.

**Validation:** Kalkidan Gezahegn, Selamawit Semagn, Mohammed Feyisso Shaka.

**Visualization:** Kalkidan Gezahegn, Selamawit Semagn, Mohammed Feyisso Shaka.

**Writing – original draft:** Kalkidan Gezahegn, Selamawit Semagn, Mohammed Feyisso Shaka.

**Writing – review & editing:** Kalkidan Gezahegn, Mohammed Feyisso Shaka.

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
