## [Decision Letter · Decision Letter 0]

9 Nov 2020

PONE-D-20-12624

Prevalence of sexual violence and its associated factors among housemaids attending evening schools in urban setups of Gedeo zone, Southern Ethiopia: A school based cross sectional study

PLOS ONE

Dear Dr. Shaka,

Thank you for submitting your manuscript to PLOS ONE. After careful consideration, we feel that it has merit but does not fully meet PLOS ONE’s publication criteria as it currently stands. Therefore, we invite you to submit a revised version of the manuscript that addresses the points raised during the review process.

We look forward to receiving your revised manuscript.

Kind regards,

Nülüfer Erbil, Ph.D, Prof.

Academic Editor

PLOS ONE

Journal Requirements:

2. Please address the following:

- Please include additional information regarding the interview guide used in the study and ensure that you have provided sufficient details that others could replicate the analyses. For instance, if you developed a guide as part of this study and it is not under a copyright more restrictive than CC-BY, please include a copy, in both the original language and English, as Supporting Information. In addition, please include the details of the pre-testing of this tool, i.e. the number of participants and where they were recruited from.

- Please refrain from stating p values as 0.00, either report the exact value or employ the format p<0.001.

3. You indicated that you had ethical approval for your study. In your Methods section, please ensure you have also stated whether you obtained consent from parents or guardians of the minors included in the study or whether the research ethics committee or IRB specifically waived the need for their consent.

6. Please ensure that you refer to Figure 1 in your text as, if accepted, production will need this reference to link the reader to the figure.

7. We note you have included a table to which you do not refer in the text of your manuscript. Please ensure that you refer to Table 1 and Table 2 in your text; if accepted, production will need this reference to link the reader to the Table.

8. Please include a copy of Table 3 which you refer to in your text on page 6.

Reviewers' comments:

Reviewer's Responses to Questions

**Comments to the Author**

1. Is the manuscript technically sound, and do the data support the conclusions?

Reviewer #1: Yes

Reviewer #2: Yes

2. Has the statistical analysis been performed appropriately and rigorously? 

Reviewer #1: Yes

Reviewer #2: Yes

3. Have the authors made all data underlying the findings in their manuscript fully available?

Reviewer #1: Yes

Reviewer #2: No

4. Is the manuscript presented in an intelligible fashion and written in standard English?

Reviewer #1: No

Reviewer #2: No

5. Review Comments to the Author

Reviewer #1: Many expressions of the author are not standard, including the display of the table. For example, when the author expresses the result, sometimes the numbers are expressed in Arabic numerals, sometimes in words, and some sentences are not clear.Please refer to the published article format and expression for revision.

Reviewer #2: Does the manuscript adhere to the PLOS Data Policy? Additional details can be found at http://journals.plos.org/plosone/s/materials-and-software-sharing. (Answer options: Yes, No

"I don’t know"

1. It is obvious that the writing need to be improved. I suggest the authors ask for a help from someone whose native language is English.

2. Section of “sample technique”:

1) The author described that a random cluster sampling was used to choose the evening school and then assigned the study participants according to the class grades. We didn’t find the class distribution of participants. Among the participants, “279 (70.8%) were adolescents (15-19 years old) whereas 115 (29.2%) were young adults (20- 24 years)”. Does the unbalanced sample exist?

2) “random sampling technique was used”, “data collectors were revisited the class room two times at different time intervals and for those who were absent after two revisits, the participant was replaced by another in the sampling frame”. How many women were replaced? Does the replace strategy have any effects on the study results?

3. Section of “measurement”

1) The sexual harassment is the most common of sexual violence experience for housemaid have experienced in this study. The author didn’t explain sexual harassment in detail, has one appropriate joke related to sex been included? If this, the proportion of sexual violence might be overestimated.

2) The alcoholic use has not been evaluated semi-quantitatively or quantitatively. This is a common-sense question: What time and how much of the alcohols consumption is the most important for the victim suffer from sexual violence.

4. Section of “results”:

1) In the descriptive analysis section, the author only needs to reveal the proportion of some important variables, for example those covariates lead to significant changes in the logistic model or other characteristic variables related to sexual violent founded from previous study.

2) Are the forms of the tables’ correct? Please refer to the author guidance of PLOS ONE.

5. Section of “Discussion”

1) The author discussed the results in comparability instead of analysis.

2) “This might be due to…”, “The possible explanation for this finding…”, “The possible suggestion might be…”, “This result might be due to …” were used by authors to explain the association between the related factors and sexual violence. It is suspicious that the author’s knowledge for the study field or the amounts of reference the author has read.

3) It is interesting for the findings that the educational attainments of fathers were associated with the sexual violence for the housemaids suffered. A health education system should be considered to help establishing an intimate father-daughter relationship in these districts.

6. PLOS authors have the option to publish the peer review history of their article (what does this mean?). If published, this will include your full peer review and any attached files.

Reviewer #1: No

Reviewer #2: No

---

## [Author Response · Author response to Decision Letter 0]

7 Jan 2021

Rebuttal Letter_Response to reviewers

PONE-D-20-12624

Prevalence of sexual violence and its associated factors among housemaids attending evening schools in urban setups of Gedeo zone, Southern Ethiopia: A school based cross sectional study

PLOS ONE

2. Please address the following:

- Please include additional information regarding the interview guide used in the study and ensure that you have provided sufficient details that others could replicate the analyses. For instance, if you developed a guide as part of this study and it is not under a copyright more restrictive than CC-BY, please include a copy, in both the original language and English, as Supporting Information. In addition, please include the details of the pre-testing of this tool, i.e. the number of participants and where they were recruited from.

- Please refrain from stating p values as 0.00, either report the exact value or employ the format p<0.001.

 Response: The detail about pretest was explained in the manuscript page four paragraph one. Additional information about the tool was included and due to copyright issue we couldn’t include as supporting information. P-value was corrected accordingly.

3. You indicated that you had ethical approval for your study. In your Methods section, please ensure you have also stated whether you obtained consent from parents or guardians of the minors included in the study or whether the research ethics committee or IRB specifically waived the need for their consent.

 Response: It is explained in the manuscript under Ethical consideration section, page 6, Paragraph 1.

 Response: We have uploaded the data on Dryad with DOI of https://doi.org/10.5061/dryad.hhmgqnkfv

 Response: Corrected accordingly. Page 6 paragraph 1

6. Please ensure that you refer to Figure 1 in your text as, if accepted, production will need this reference to link the reader to the figure.

 Response: Corrected. Page 8

7. We note you have included a table to which you do not refer in the text of your manuscript. Please ensure that you refer to Table 1 and Table 2 in your text; if accepted, production will need this reference to link the reader to the Table.

 Response: We have corrected the typing error regarding the table numbers

8. Please include a copy of Table 3 which you refer to in your text on page 6.

Reviewers' comments:

Reviewer's Responses to Questions

Comments to the Author

Reviewer #1: Many expressions of the author are not standard, including the display of the table. For example, when the author expresses the result, sometimes the numbers are expressed in Arabic numerals, sometimes in words, and some sentences are not clear. Please refer to the published article format and expression for revision.

Response: We have tried to make the presentation of the table to be to the standard. Unnecessary, numerical presentations in the table were modified 

Reviewer #2: Does the manuscript adhere to the PLOS Data Policy? Additional details can be found at http://journals.plos.org/plosone/s/materials-and-software-sharing. (Answer options: Yes, No

"I don’t know"

1. It is obvious that the writing need to be improved. I suggest the authors ask for a help from someone whose native language is English.

Response: The manuscript English writing was proofread by Proffessional proofreaders to improve readability. 

2. Section of “sample technique”:

1) The author described that a random cluster sampling was used to choose the evening school and then assigned the study participants according to the class grades. We didn’t find the class distribution of participants. Among the participants, “279 (70.8%) were adolescents (15-19 years old) whereas 115 (29.2%) were young adults (20- 24 years)”. Does the unbalanced sample exist

Response: Class distribution of the students was briefly explained in the methodology under sampling technique subsection page 4. Regarding issue about existence of unbalanced sampling, the proportion is not due to unbalanced sampling. It entirely due to the actual age distribution of the population.

2) “random sampling technique was used”, “data collectors were revisited the class room two times at different time intervals and for those who were absent after two revisits, the participant was replaced by another in the sampling frame”. How many women were replaced? Does the replace strategy have any effects on the study results? 

Response: The number of women who replaced for their absence was not significant where only three women were missed and replaced. Replacement was also made by randomly selecting the participant from the previously unselected eligible population.

3. Section of “measurement”

1) The sexual harassment is the most common of sexual violence experience for housemaid have experienced in this study. The author didn’t explain sexual harassment in detail, has one appropriate joke related to sex been included? If this, the proportion of sexual violence might be overestimated.

Response: Explained under measurement subsection second paragraph page 5

2) The alcoholic use has not been evaluated semi-quantitatively or quantitatively. This is a common-sense question: What time and how much of the alcohols consumption is the most important for the victim suffer from sexual violence. 

Response: The ASSIST tool is standard tool for assessment of the common substance uses based on the frequency and duration of use. We used that tool. With that tool we couldn’t find any women fulfilling criteria for substance use. So we measure the episodes of occasional use and classified alcohol use as ever use or not at all 

4. Section of “results”:

1) In the descriptive analysis section, the author only needs to reveal the proportion of some important variables, for example those covariates lead to significant changes in the logistic model or other characteristic variables related to sexual violent founded from previous study. 

Response: We have tried to remove some variables thought to be less important to present in this study. However, as one of the objective of this study is to describe the violence and the characteristics of the study participants, we retained most of the variables thought to have relevant descriptive information. 

2) Are the forms of the tables’ correct? Please refer to the author guidance of PLOS ONE.

5. Section of “Discussion”

1) The author discussed the results in comparability instead of analysis.

Response: We have tried to present with the analysis of the findings

2) “This might be due to…”, “The possible explanation for this finding…”, “The possible suggestion might be…”, “This result might be due to …” were used by authors to explain the association between the related factors and sexual violence. It is suspicious that the author’s knowledge for the study field or the amounts of reference the author has read.

Response: We have modified our way of explanation accordingly

3) It is interesting for the findings that the educational attainments of fathers were associated with the sexual violence for the housemaids suffered. A health education system should be considered to help establishing an intimate father-daughter relationship in these districts. 

Response: The finding of the study was disseminated to the local health planners and this will be an input to address the issue.

Thank you for your interesting and valuable comments that made us to improve our work substantially.

---

## [Decision Letter · Decision Letter 1]

10 Jun 2021

PONE-D-20-12624R1

Prevalence of sexual violence and its associated factors among housemaids attending evening schools in urban settings of Gedeo zone, Southern Ethiopia: A school based cross sectional study

PLOS ONE

Dear Dr. Shaka,

Thank you for submitting your manuscript to PLOS ONE. After careful consideration, we feel that it has merit but does not fully meet PLOS ONE’s publication criteria as it currently stands. Therefore, we invite you to submit a revised version of the manuscript that addresses the points raised during the review process.

We look forward to receiving your revised manuscript.

Kind regards,

Nülüfer Erbil, Ph.D, Prof.

Academic Editor

PLOS ONE

Additional Editor Comments:

Reviewers' comments:

Reviewer's Responses to Questions

**Comments to the Author**

1. If the authors have adequately addressed your comments raised in a previous round of review and you feel that this manuscript is now acceptable for publication, you may indicate that here to bypass the “Comments to the Author” section, enter your conflict of interest statement in the “Confidential to Editor” section, and submit your "Accept" recommendation.

Reviewer #2: All comments have been addressed

Reviewer #3: All comments have been addressed

2. Is the manuscript technically sound, and do the data support the conclusions?

Reviewer #2: Partly

Reviewer #3: Yes

3. Has the statistical analysis been performed appropriately and rigorously? 

Reviewer #2: Yes

Reviewer #3: Yes

4. Have the authors made all data underlying the findings in their manuscript fully available?

Reviewer #2: Yes

Reviewer #3: Yes

5. Is the manuscript presented in an intelligible fashion and written in standard English?

Reviewer #2: Yes

Reviewer #3: Yes

6. Review Comments to the Author

Reviewer #2: The authours have addressed my previous comments. I am satisfated with their hard working on the field.

Reviewer #3: I think, this study is an important addition to the literatüre, but I have some minor concerns.

Data collection

The authors used the face to face interview how did the participants feel comfortable such a sensitive issue? How did they prevent the bias and protect the correct of the answer?

why did the author create the survey English language? Are these instruments available in their language, if not, did you make content and language validity of them?

Ethical Consideration

The main concern has protected minors, how did you protect them, when they disclosure sexual violence. What is your legal responsibility for the researcher? Is it any mandatory report about disclosure about SV. Please give your legal policy about this concern..

Discussion

some findings haven't discussed, if they don't discuss, you should eliminate some findings based on your discussion.

Please give the broader literature by supporting international and national studies.

Please give some other references.

for the international audience please give the other global studies results too.

the whole discussion should support the international and national references

some findings haven't discussed, if they don't discuss, you should eliminate some findings based on your discussion.

The main limitation in this study seems that it was neglected to child protection

I found the other review changes satisfactory for manuscript .

7. PLOS authors have the option to publish the peer review history of their article (what does this mean?). If published, this will include your full peer review and any attached files.

Reviewer #2: No

Reviewer #3: No

---

## [Author Response · Author response to Decision Letter 1]

5 Sep 2021

Response to reviewers

• The authors used the face to face interview how did the participants feel comfortable such a sensitive issue? How did they prevent the bias and protect the correct of the answer?

o From the report of the data collectors during the time of pre-test, the feeling of the respondents for the questions was not embarrassing and the respondents were very comfortable with the approach. The same was reported form the daily routine follow up of actual data collection 

o The data collectors were well trained to handle such issue

o The participants were adequately reassured for the confidentiality of the information they provide

o The interview was conducted at isolated place with adequate privacy

• Why did the author create the survey English language? Are these instruments available in their language, if not, did you make content and language validity of them? 

We have used the survey tool that was used previously in comparable settings. We couldn’t find the tool in local language. We have tried to familiarize it with the context through pre-test and training of the surveyors. 

• Ethical Consideration

The main concern has protected minors, how did you protect them, when they disclosure sexual violence. What is your legal responsibility for the researcher? Is it any mandatory report about disclosure about SV. Please give your legal policy about this concern. 

Confidentiality of the response was highly secured. As a researcher, we have provided contact information for any possible risk associated with the disclosure information during data collection. The respondents were advised for possible legal body and regulation regarding sexual violence according to guiding principle of acting in the best interests of the child. Otherwise, the right to lodge a complaint shall be exercised only by legal representative according to regulation in the locality and most of the participants of our study were older children (>15 years) and there were no issues regarding capacity to consent or a relationship of power as of our data collectors report from day to day follow up.

Discussion

• We have revised the discussion according to your suggestion including discussion of findings that were missed in the previous revision 

Thank you for your constructive comments. We are really appreciate the contribution you made to the improvement of our work.

Mohammed Feyisso Shaka

Kalkidan Gezahagn

Selamawith Semagn

---

## [Decision Letter · Decision Letter 2]

11 Oct 2021

Prevalence of sexual violence and its associated factors among housemaids attending evening schools in urban settings of Gedeo zone, Southern Ethiopia: A school based cross sectional study

PONE-D-20-12624R2

Dear Dr. Shaka,

We’re pleased to inform you that your manuscript has been judged scientifically suitable for publication and will be formally accepted for publication once it meets all outstanding technical requirements.

Kind regards,

Nülüfer Erbil, Ph.D, Prof.

Academic Editor

PLOS ONE

---

## [Editor Report · Acceptance letter]

21 Oct 2021

PONE-D-20-12624R2 

Prevalence of sexual violence and its associated factors among housemaids attending evening schools in urban settings of Gedeo zone, Southern Ethiopia: A school based cross sectional study 

Dear Dr. Shaka:

I'm pleased to inform you that your manuscript has been deemed suitable for publication in PLOS ONE. Congratulations! Your manuscript is now with our production department. 

Kind regards, 

on behalf of

Dr. Nülüfer Erbil 

Academic Editor

PLOS ONE